# Task-Agnostic and Adaptive-Size BERT Compression

## Abstract

While pre-trained language models such as BERT and RoBERTa have achieved impressive results on various natural language processing tasks, they have huge numbers of parameters and suffer from huge computational and memory costs, which make them difficult for real-world deployment. Hence, model compression should be performed in order to reduce the computation and memory cost of pre-trained models. In this work, we aim to compress BERT and address the following two challenging practical issues: (1) The compression algorithm should be able to output multiple compressed models with different sizes and latencies, so as to support devices with different kinds of memory and latency limitations; (2) the algorithm should be downstream task agnostic, so that the compressed models are generally applicable for different downstream tasks. We leverage techniques in neural architecture search (NAS) and propose *NAS-BERT*, an efficient method for BERT compression. NAS-BERT trains a big supernet on a carefully designed search space containing various architectures and outputs multiple compressed models with adaptive sizes and latency. Furthermore, the training of NAS-BERT is conducted on standard self-supervised pre-training tasks (e.g., masked language model) and does not depend on specific downstream tasks. Thus, the models it produces can be used across various downstream tasks. The technical challenge of NAS-BERT is that training a big supernet on the pre-training task is extremely costly. We employ several techniques including block-wise search, search space pruning, and performance approximation to improve search efficiency and accuracy. Extensive experiments on GLUE benchmark datasets demonstrate that NAS-BERT can find lightweight models with better accuracy than previous approaches, and can be directly applied to different downstream tasks with adaptive model sizes for different requirements of memory or latency.

## 1 Introduction

Pre-trained Transformer (Vaswani et al., 2017)-based language models like BERT (Devlin et al., 2019), XLNet (Yang et al., 2019) and RoBERTa (Liu et al., 2019) have achieved impressive performance on a variety of downstream natural language processing tasks. These models are pre-trained on massive language corpus via self-supervised tasks to learn language representation and fine-tuned on specific downstream tasks. Despite their effectiveness, these models are quite expensive in terms of computation and memory cost, which makes them difficult for the deployment on different downstream tasks and various resource-restricted scenarios such as online servers, mobile phones, and embedded devices. Therefore, it is crucial to compress pre-trained models for practical deployment.

Recently, a variety of compression techniques have been adopted to compress pre-trained models, such as pruning (McCarley, 2019; Gordon et al., 2020), weight factorization (Lan et al., 2019), quantization (Shen et al., 2020; Zafrir et al., 2019), and knowledge distillation (Sun et al., 2019; Sanh et al., 2019; Chen et al., 2020; Jiao et al., 2019; Hou et al., 2020; Song et al., 2020). Several existing works (Tsai et al., 2020; McCarley, 2019; Gordon et al., 2020; Sanh et al., 2019; Zafrir et al., 2019; Chen et al., 2020; Lan et al., 2019; Sun et al., 2019) compress a large pre-trained model into a small or fast model with fixed size on the pre-training or fine-tuning stage and have achieved good compression efficiency and accuracy. However, from the perspective of practical deployment, a fixed size model cannot be deployed in devices with different memory and latency constraints. For example, smaller models are preferred in embedded devices than in online servers, and faster

inference speed is more critical in online services than in offline services. On the other hand, some previous methods (Chen et al., 2020; Hou et al., 2020) compress the models on the fine-tuning stage for each specific downstream task. This can achieve good accuracy due to the dedicated design in each task. However, compressing the model for each task can be laborious and a compressed model for one task may not generalize well on another downstream task.

In this paper, we study the BERT compression in a different setting: the compressed models need to cover different sizes and latencies, in order to support devices with different kinds of memory and latency constraints; the compression is conducted on the pre-training stage so as to be downstream task agnostic. To this end, we propose NAS-BERT, which leverages neural architecture search (NAS) to automatically compress BERT models. We carefully design a search space that contains multi-head attention (Vaswani et al., 2017), separable convolution (Kaiser et al., 2018), feed-forward network and identity operations with different hidden sizes to find efficient models. In order to search models with adaptive sizes that satisfy diverse requirements of memory and latency constraints in different devices, we train a big supernet that contains all the candidate operations and architectures with weight sharing (Bender et al., 2018; Cai et al., 2018; 2019; Yu et al., 2020). In order to reduce the laborious compressing on each downstream task, we directly train the big supernet and get the compressed model on the pre-training task to make it applicable across different downstream tasks.

However, it is extremely expensive to directly perform architecture search in a big supernet on the heavy pre-training task. To improve the search efficiency and accuracy, we employ several techniques including block-wise search, search space pruning and performance approximation during the search process: (1) We adopt block-wise search (Li et al., 2020a) to divide the supernet into blocks so that the size of the search space can be reduced exponentially. To train each block, we leverage a pre-trained teacher model, divide the teacher model into blocks similarly, and use the input and output hidden states of the corresponding teacher block as paired data for training. (2) To further reduce the search cost of each block (even if block-wise search has greatly reduced the search space) due to the heavy burden of the pre-training task, we propose progressive shrinking to dynamically prune the search space according to the validation loss during training. To ensure that architectures with different sizes and latencies can be retained during the pruning process, we further divide all the architectures in each block into several bins according to their model sizes and perform progressive shrinking in each bin. (3) We obtain the compressed models under specific constraints of memory and latency by assembling the architectures in every block using performance approximation, which can reduce the cost in model selection.

We evaluate the models compressed by NAS-BERT on the GLUE benchmark (Wang et al., 2018). The results show that NAS-BERT can find lightweight models with various sizes from 5M to 60M with better accuracy than that achieved by previous work. Our contributions of NAS-BERT can be summarized as follows:

- We carefully design a search space that contains various architectures and different sizes, and apply NAS on the pre-training task to search for efficient lightweight models, which is able to deliver adaptive model sizes given different requirements of memory or latency and apply for different downstream tasks.

- We further apply block-wise search, progressive shrinking and performance approximation to reduce the huge search cost and improve the search accuracy.

- Experiments on the GLUE benchmark datasets demonstrate the effectiveness of NAS-BERT for model compression.

## 2 RELATED WORK

**BERT Model Compression**  Recently, compressing pre-trained language models has been studied extensively and several techniques have been proposed such as knowledge distillation, pruning, weight factorization, quantization and so on. Existing works (Tsai et al., 2020; Sanh et al., 2019; Sun et al., 2019; Song et al., 2020; Jiao et al., 2019; Lan et al., 2019; Zafrir et al., 2019; Shen et al., 2020; Wang et al., 2019b; Lan et al., 2019; Zafrir et al., 2019; Chen et al., 2020) aim to compress the pre-trained model into a fixed size of the model and have achieved a trade-off between the small parameter size (usually no more than 66M) and the good performance. However, these compressed models cannot be deployed in devices with different memory and latency constraints.

Recent works (Hou et al., 2020) can deliver adaptive models for each specific downstream task and demonstrate the effectiveness of the task-oriented compression. For practical deployment, it can be laborious to compress models from each task. Other works (Fan et al., 2019) can produce compressed models on the pre-training stage that can directly generalize to downstream tasks, and allow for efficient pruning at inference time. However, they do not explore the potential of different architectures as in our work. Different from existing works, NAS-BERT aims for task-agnostic compression on the pre-training stage which eliminates the laborious compression for each specific downstream task, and carefully designs the search space which is capable to explore the potential of different architectures and deliver various models given diverse memory and latency requirements.

**Neural Architecture Search for Efficient Models**   Many works have leveraged NAS to search efficient models (Liu et al., 2018; Cai et al., 2018; Howard et al., 2019; Tan & Le, 2019; Cai et al., 2019; Yu et al., 2020; Wang et al., 2020a; Tsai et al., 2020). Most of them focus on computer vision tasks and rely on specific designs on the convolutional layers (e.g., inverted bottleneck convolution (Howard et al., 2019) or elastic kernel size (Cai et al., 2019; Yu et al., 2020)). Among them, once-for-all (Cai et al., 2019) and BigNAS (Yu et al., 2020) train a big supernet that contains all the candidate architectures and can get a specialized sub-network by selecting from the supernet to support various requirements (e.g., model size and latency). HAT (Wang et al., 2020a) also trains a supernet with the adaptive widths and depths for machine translation tasks. Our proposed NAS-BERT also trains a big supernet. However, different from these methods, we target model compression for BERT at the pre-training stage, which is a more challenging task due to the large model size and huge pre-training cost. Therefore, we introduce several techniques including block-wise search, progressive shrinking, and performance approximation to reduce the training cost and improve search efficiency. Tsai et al. (2020) apply one-shot NAS to search a faster Transformer but they cannot deliver multiple architectures to meet various constraints for deployment. Different from Tsai et al. (2020), NAS-BERT 1) progressively shrinks the search space to allocate more resources to promising architectures and thus can deliver various architectures without adding much computation; 2) designs bins in the shrinking algorithm to guarantee that we can search architectures to meet diverse memory and latency constraints. 3) explores novel architectures with convolution layer, multi-head attention, and feed-forward layer, and achieves better performance than previous works for BERT compression.

## 3   METHOD

In this section, we describe NAS-BERT, which conducts neural architecture search to find small, novel and accurate BERT models. To meet the requirements of deployment for different memory and latency constraints and across different downstream tasks, we 1) train a supernet with a novel search space that contains different sizes of models for various resource-restricted devices, and 2) directly search the models on the pre-training task to make them generalizable on different downstream tasks. The method can be divided into three steps: 1) search space design (Section 3.1); 2) supernet training (Section 3.2); 3) model selection (Section 3.3). Due to the huge cost to train the big supernet on the heavy pre-training task and select compressed models under specific constraints, we introduce several techniques including block-wise search, search space pruning and performance approximation in Section 3.2 and 3.3 to reduce the search space and improve the search efficiency.

### 3.1   SEARCH SPACE DESIGN

A novel search space allows the potential of combinations of different operations, instead of simply stacking basic Transformer block (multi-head attention and feed-forward network) as in the original BERT model. We adopt the chain-structured search space (Elsken et al., 2018), and construct an over-parameterized supernet $\mathcal{A}$ with $L$ layers and each layer contains all candidate operations in $\mathcal{O} = \{o_1, \cdots, o_C\}$, where $C$ is the number of predefined candidate operations. Residual connection is applied to each layer by adding the input to the output. There are $C^L$ possible paths (architectures) in the supernet, and a specific architecture $a = (a^1, \cdots, a^L)$ is a sub-net (path) in the supernet, where $a^l \in \mathcal{O}$ is the operation in the $l$-th layer, as shown in Fig. 2 (a). We adopt weight sharing mechanism that is widely used in NAS (Bender et al., 2018; Cai et al., 2019) for efficient training, where each architecture (path) shares the same set of operations in each layer.

We further describe each operation in $\mathcal{O}$ as follows: 1) Multi-head attention (MHA) and feed-forward network (FFN), which are the two basic operations in Transformer and are popular in pre-training models (in this way we can cover BERT model as a subnet in our supernet). 2) Separable convolution (SepConv), whose effectiveness and efficiency in natural language processing tasks have been demonstrated by previous work (Kaiser et al., 2018; Karatzoglou et al., 2020). 3) Identity operation, which can support architectures with the number of layers less than $L$. Identity operation is regarded as a placeholder in a candidate architecture and can be removed to obtain a shallower network. More detailed considerations on choosing the operation set are in Appendix A.1. Apart from different types of operations, to allow adaptive model sizes, each operation can have different hidden sizes: {128, 192, 256, 384, 512}. In this way, architectures in the search space can be of different depths and widths. The complete candidate operation set $\mathcal{O}$ contains $(1+1+3)*5+1 = 26$ operations, where the first product term represents the number of types of operations and 3 represents the SepConv with different kernel size {3, 5, 7}, the second product term represents that there are 5 different hidden sizes. We list 26 operations in Table 1. The detailed structure of separable convolution is shown in Fig. 1.

| Hidden Size | 128 | 192 | 256 | 384 | 512 |
|---|---|---|---|---|---|
| MHA | 2 Heads | 3 Heads | 4 Heads | 6 Heads | 8 Heads |
| FFN | 512 | 768 | 1024 | 1536 | 2048 |
| SepConv | Kernel 3 Kernel 5 Kernel 7 | Kernel 3 Kernel 5 Kernel 7 | Kernel 3 Kernel 5 Kernel 7 | Kernel 3 Kernel 5 Kernel 7 | Kernel 3 Kernel 5 Kernel 7 |
| Identity | Identity | | | | |

Table 1: Candidate operation set. For each type of operation including multi-head attention (MHA), feed-forward network (FFN) and separable convolution (SepConv) in each row, we list the number of heads in MHA and the size of the intermediate layer in FFN, and kernel size in SepConv under different hidden sizes (in different columns).

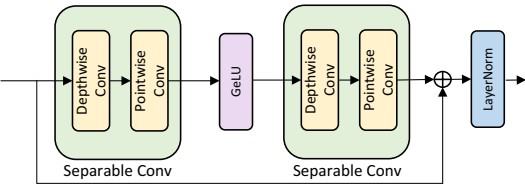

Figure 1: Structure of separable convolution.

## 3.2 SUPERNET TRAINING

### 3.2.1 BLOCK-WISE TRAINING WITH KNOWLEDGE DISTILLATION

Directly training the whole supernet causes huge cost due to its large model size and huge search space. With limited computational resources (total training time, steps, etc.), the amortized training time of each architecture from the huge search space is insufficient for accurate evaluation (Chu et al., 2019; Luo et al., 2019; Li et al., 2020b). Inspired by Li et al. (2020a), we adopt block-wise search to uniformly divide the supernet $\mathcal{A}$ into $N$ blocks $(\mathcal{A}_1, \mathcal{A}_2, \cdots, \mathcal{A}_N)$ to reduce the search space and improve the efficiency. To train each block independently and effectively, knowledge distillation is applied with a pre-trained BERT model. The pre-trained BERT model (teacher) is divided into corresponding $N$ blocks as in Fig. 2. The input and output hidden states of the corresponding block in the teacher model are used as the paired data to train the block in the supernet (student). Specifically, the $n$-th student block receives the output of the $(n-1)$-th teacher block as the input and is optimized to predict the output of the $n$-th teacher block with mean square loss

$$\mathcal{L}_n = ||f(\mathbf{Y}_{n-1}; \mathcal{A}_n) - \mathbf{Y}_n||_2^2, \tag{1}$$

where $f(\cdot; \mathcal{A}_n)$ is the mapping function of $n$-th block $\mathcal{A}_n$, $\mathbf{Y}_n$ is the output of the $n$-th block of the teacher model ($\mathbf{Y}_0$ is the output of the embedding layer of the teacher model). At each training step,

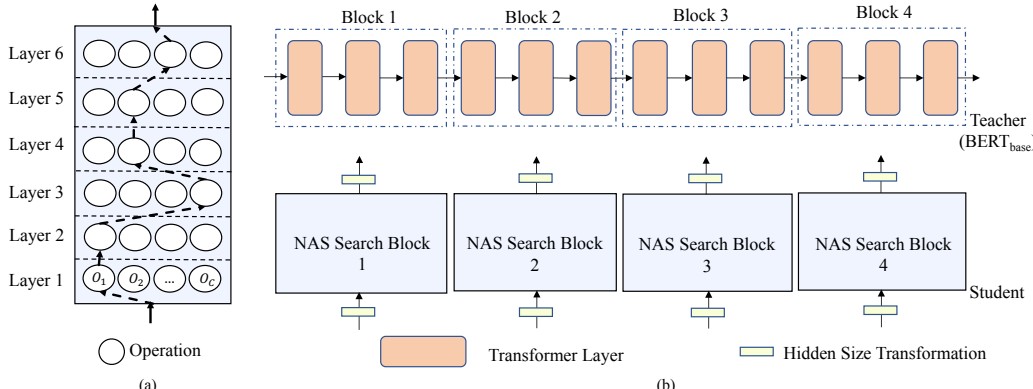

Figure 2: (a) an architecture (path) in the supernet. (b) an illustration of block-wise distillation ($N = 4$ blocks). The supernet (student) and the pre-trained teacher model are divided into blocks respectively and each student block is trained to mimic the corresponding teacher block.

we randomly sample an architecture from the search space following Guo et al. (2019); Bender et al. (2018); Cai et al. (2019), which is memory-efficient due to the single path optimization. Different from Li et al. (2020a), we allow different hidden sizes and incorporate identity layer in each block to support elastic width and depth to derive models that meet various requirements. Besides, the search space within each block in our work is larger compared to Li et al. (2020a) (100x larger) which is much more sample in-efficient and requires more techniques (described in Section 3.2.2) to improve the training efficiency. Since the hidden sizes of the student block may be different from that in the teacher block, we cannot directly leverage the input and output hidden of the teacher block as the training data of the student block. To solve this problem, we use a learnable linear transformation layer at the input and output of student block to transform each hidden size to match that of the teacher block, as shown in Fig. 2.

### 3.2.2 PROGRESSIVE SHRINKING

Although block-wise training can largely reduce the search space, the supernet still requires huge time for convergence due to the heavy pre-training task. To further improve the training effectiveness and efficiency, we propose to progressively shrink the search space in each block during training to allocate more training resources to more promising candidates (Wang et al., 2019a; Li et al., 2020b; Luo et al., 2020). However, simply pruning the architectures cannot ensure to obtain different sizes of models, since larger models in $\mathcal{A}_n$ are likely to be pruned on the early stage of training due to its difficulty of optimization (Chu et al., 2019; Luo et al., 2019; Li et al., 2020b) and smaller models are likely to be pruned at the late stage due to limited capacity. Therefore, we assign the architectures in $\mathcal{A}_n$ into different *bins* where each bin represents a short range of model sizes. Besides, we also apply latency constraints in each bin to avoid models accepted parameter size but large latency. Denote $p_b = \frac{b}{B} \cdot p(a^t)$ and $l_b = \frac{b}{B} \cdot l(a^t)$ as the maximum parameter size and latency for the $b$-th bin, where $p(\cdot)$ and $l(\cdot)$ calculate the parameter size and latency, $a^t$ is the largest model in the search space and $B$ is the number of bins. The architecture $a$ in $b$-th bin should meet (1) $p_b > p(a) > p_{b-1}$ and (2) $l_b > l(a) > l_{b-1}$. Architectures that cannot satisfy the constraint of latency are removed.

Then we conduct the progressive shrinking algorithm in each bin at the end of each training epoch as follows: 1) **Sample** $E$ architectures in each bin and get the validation losses on the dev set; 2) **Rank** the $E$ architectures according to their validation losses in descending order; 3) **Remove** $R$ architectures with the largest losses. The shrinking algorithm terminates when there are only $m$ architectures left in the search space to avoid all the architectures being deleted. The design of bins ensures the diversity of models when shrinking the search space, which makes it possible to select a model given diverse constraints at the model selection stage.

### 3.3 MODEL SELECTION

After the supernet training with progressive shrinking, each block contains $m * B$ possible archi-tectures and the whole supernet ($N$ blocks) contains $(m * B)^N$ complete architectures. The model selection procedure is as follows: 1) We build a large lookup table $LT_{arch}$ with $(m * B)^N$ items,

where each item contains the meta-information of a complete architecture: *(architecture, parameter, latency, loss)*. Since it is extremely time-consuming to measure the exact latency and loss for $(m * B)^N$ (e.g., $10^8$ in our experiments) architectures, we use performance approximation to obtain the two values as described in the next paragraph. 2) For a given constraint of model size and inference latency, we select the top $T$ architectures with low loss from $LT_{arch}$ that meet the parameter and latency constraint. 3) We evaluate the validation loss of the top $T$ complete architectures on the dev set and select the best one as the final compressed model. The compressed model is associated with an embedding layer with adaptive size rather than a fixed size, and the embedding size is determined by the rules according to the requirement of target model size (see Appendix A.5).

Next we introduce the performance approximation of the latency and loss when building the lookup table $LT_{arch}$. We measure the latency of each candidate operation (just 26 in our design) on the target device and store in a lookup table $LT_{lat}$ in advance, and then approximate the latency of an architecture $l(a)$ by $l(a) = \sum_{l=1}^{L} l(a^l)$ following Cai et al. (2018), where $l(a^l)$ is from $LT_{lat}$. To approximate the loss of an architectures in $LT_{arch}$, we add up the block-wise distillation loss of the sub-architecture in each block on the dev set. Obtaining the dev loss of all sub-architectures in all blocks only involves $m * B * N$ evaluations.

## 4 EXPERIMENT

### 4.1 EXPERIMENTAL SETUP

**Supernet Training Setup** The supernet consists of $L = 24$ layers, which is consistent with BERT$_{base}$ (Devlin et al., 2019) (BERT$_{base}$ has 12 Transformer layers with 24 sub-layers in total, since each Transformer layer has a MHA and FFN). We use a pre-trained BERT$_{base}$ (Devlin et al., 2019) as the teacher model. The detailed configurations of the search space and teacher model training can be found in Appendix A.1 and Appendix A.2. The supernet is divided into $N = 4$ blocks and the search space in each block is divided into $B = 10$ bins. We first train the supernet for 3 epochs without progressive shrinking as a warm start, and then begin to shrink at the end of each later epoch. We randomly sample $E = 2000$ architectures for validation (evaluate all architectures when the number of architectures in search space is less than 2000) and perform the progressive shrinking to remove $R = E/2$ architectures for each bin as in Section 3.2.2. The shrinking process terminates when only $m = 10$ architectures are left in each bin in each block, and the training also ends. The considerations about how to decide these hyper-parameters are described in Appendix A.5. The supernet is trained on English Wikipedia plus BookCorpus (16GB size), with a batch size of 1024 sentences and each sentence consists of 512 tokens. The training costs 3 days on 32 NVIDIA P40 GPUs while training the BERT$_{base}$ teacher model costs 5 days. Other training configurations remain the same as the teacher model. The latency used in progressive shrinking and model selection is measured on Intel(R) Xeon(R) CPU E5-2690 v4 @ 2.60 GHz with 12 cores, but our method can be easily applied to other devices (e.g., mobile platforms, embedded devices) by using the lookup table $LT_{lat}$ (described in Section 3.3) measured and built for the corresponding device. We select $T = 100$ models from the table $LT_{arch}$ on the model selection stage. In progressive shrinking, to reduce the time of evaluating all the candidates, we only evaluate on 5 batches rather than the whole dev set, which is accurate enough for the pruning according to our preliminary study.

**Evaluation Setup on Downstream Tasks** We evaluate the effectiveness of NAS-BERT by pre-training the compressed models on the original pre-training task and fine-tuning on the GLUE benchmark (Wang et al., 2018), which includes CoLA (Warstadt et al., 2018), SST-2 (Socher et al., 2013), MRPC (Dolan & Brockett, 2005), STS-B (Cer et al., 2017), QQP (Chen et al., 2018), MNLI (Williams et al., 2018), QNLI (Rajpurkar et al., 2016) and RTE (Dagan et al., 2006). Similar to previous methods (Sanh et al., 2019; Wang et al., 2020b; Turc et al., 2019; Hou et al., 2020), we also apply knowledge distillation and conduct it on two stages (i.e., pre-training and fine-tuning) as the default setting for evaluation. The details of two-stage distillation can be found in Appendix A.4. Considering the focus of our work is to compress pre-trained models with novel architectures instead of knowledge distillation, we only use prediction layer distillation and leave the various distillation techniques like layer-by-layer distillation, embedding layer distillation and attention matrix distillation (Sun et al., 2019; Jiao et al., 2019; Sanh et al., 2019; Hou et al., 2020; Wang et al., 2020b) that

| Setting | FLOPs | Speedup | MNLI | QQP | QNLI | CoLA | SST-2 | STS-B | RTE | MRPC | AVG |
|---|---|---|---|---|---|---|---|---|---|---|---|
| $\text{BERT}_{60}$ + PF | 1.3e10 | 2.2× | 82.6 | 90.3 | 89.4 | 52.6 | 92.1 | 88.3 | 75.6 | 89.2 | 82.5 |
| NAS-$\text{BERT}_{60}$ + PF | 1.3e10 | 2.2× | 83.0 | 90.9 | 90.8 | 53.8 | 92.3 | 88.7 | 76.7 | 88.9 | **83.2** |
| $\text{BERT}_{60}$ + KD | 1.3e10 | 2.2× | 83.2 | 90.5 | 90.2 | 56.3 | 91.8 | 88.8 | 78.5 | 88.5 | 83.5 |
| NAS-$\text{BERT}_{60}$ + KD | 1.3e10 | 2.2× | 84.1 | 91.0 | 91.3 | 58.1 | 92.1 | 89.4 | 79.2 | 88.5 | **84.2** |
| $\text{BERT}_{30}$ + PF | 7.1e9 | 3.6 × | 80.0 | 89.6 | 87.6 | 40.6 | 90.8 | 87.7 | 73.2 | 84.1 | 79.2 |
| NAS-$\text{BERT}_{30}$ + PF | 7.0e9 | 3.6 × | 80.4 | 90.0 | 87.8 | 48.1 | 90.3 | 87.3 | 71.4 | 84.9 | **80.0** |
| $\text{BERT}_{30}$ + KD | 7.1e9 | 3.6 × | 80.8 | 89.8 | 88.7 | 44.7 | 90.5 | 87.6 | 70.3 | 85.2 | 79.7 |
| NAS-$\text{BERT}_{30}$ + KD | 7.0e9 | 3.6 × | 81.0 | 90.2 | 88.4 | 48.7 | 90.5 | 87.6 | 71.8 | 84.6 | **80.3** |
| $\text{BERT}_{10}$ + PF | 2.3e9 | 9.1 × | 74.0 | 87.6 | 84.9 | 25.7 | 88.3 | 85.3 | 64.1 | 81.9 | 74.0 |
| NAS-$\text{BERT}_{10}$ + PF | 2.3e9 | 8.7 × | 76.0 | 88.4 | 86.3 | 27.8 | 88.6 | 84.3 | 68.7 | 81.5 | **75.2** |
| $\text{BERT}_{10}$ + KD | 2.3e9 | 9.1 × | 74.4 | 87.8 | 85.7 | 32.5 | 86.6 | 85.2 | 66.9 | 77.9 | 74.6 |
| NAS-$\text{BERT}_{10}$ + KD | 2.3e9 | 8.7 × | 76.4 | 88.5 | 86.3 | 34.0 | 88.6 | 84.8 | 66.6 | 79.1 | **75.5** |
| $\text{BERT}_{5}$ + PF | 8.6e8 | 20.7 × | 67.7 | 84.1 | 80.9 | 10.4 | 81.6 | 81.1 | 62.8 | 78.6 | 68.4 |
| NAS-$\text{BERT}_{5}$ + PF | 8.7e8 | 23.6 × | 74.2 | 85.7 | 83.9 | 19.6 | 84.9 | 82.8 | 67.0 | 80.0 | **72.3** |
| $\text{BERT}_{5}$ + KD | 8.6e8 | 20.7 × | 67.9 | 83.2 | 80.6 | 12.6 | 82.8 | 81.0 | 61.9 | 78.1 | 68.5 |
| NAS-$\text{BERT}_{5}$ + KD | 8.7e8 | 23.6 × | 74.4 | 85.8 | 84.9 | 19.8 | 87.3 | 83.0 | 66.6 | 79.6 | **72.7** |

Table 2: Comparison of NAS-BERT models and hand-designed BERT models under different sizes (60M, 30M, 10M, 5M) on GLUE dev set. "PF" means pre-training and fine-tuning. "KD" means two-stage knowledge distillation. MNLI is reported in the matched set. Spearman correlation is reported for STS-B. Matthews correlation is reported for CoLA. Accuracy is reported for other tasks.

can further improve the performance as to future work. During fine-tuning on the GLUE benchmark, RTE, MRPC and STS-B are started from the model fine-tuned on MNLI following Liu et al. (2019).

## 4.2 RESULTS

**Accuracy of NAS-BERT**    While our NAS-BERT can compress models with adaptive sizes, we only show the results of compressed models with 60M, 30M, 10M and 5M parameter sizes (denoted as NAS-$\text{BERT}_{60}$, NAS-$\text{BERT}_{30}$, NAS-$\text{BERT}_{10}$ and NAS-$\text{BERT}_{5}$ respectively) on the GLUE benchmark due to the large evaluation cost, and list the model structures with different sizes in Appendix A.6. We compare our NAS-BERT models with hand-designed BERT models under the same parameter size and latency (denoted as $\text{BERT}_{60}$, $\text{BERT}_{30}$, $\text{BERT}_{10}$ and $\text{BERT}_{5}$ respectively). We follow several principles when manually designing the BERT models: we just use the original BERT structure (MHA plus FFN) and keep the parameter, latency, depth and width as close as possible to the corresponding NAS-BERT models. The detailed structures of the hand-designed BERT models are introduced in the next paragraph. To demonstrate the advantages of architectures searched by NAS-BERT, the comparisons are evaluated in two settings: 1) only with pre-training and fine-tuning (PF), and 2) pre-training and fine-tuning with two-stage knowledge distillation (KD). We measure the inference speedup of NAS-BERT compared with $\text{BERT}_{\text{base}}$ and inference FLOPs following Clark et al. (2019). The configurations are in Appendix A.3. The results are shown in Table 2, from which we can see that NAS-BERT outperforms hand-designed BERT baseline across almost all the tasks under various model sizes. Especially, the smaller the model size is, the larger gap can be observed (e.g., NAS-$\text{BERT}_{5}$ vs. $\text{BERT}_{5}$). The results show that NAS-BERT can search for efficient lightweight models that are better than Transformer based models.

**BERT Baselines**    We follow several principles when manually designing the BERT models in Table 2: 1) The size of the embedding layer is the same as that of the corresponding NAS-BERT model; 2) We use the original BERT structure (MHA plus FFN) and keep the parameter, latency, depth and width as close as possible to the corresponding NAS-BERT model. The baseline BERT models in Table 2 are: $\text{BERT}_{60}$ (L=10, H=512, A=8), $\text{BERT}_{30}$ (L=6, H=512, A=8), $\text{BERT}_{10}$ (L=6, H=256, A=4) and $\text{BERT}_{5}$ (L=6, H=128, A=2) where L is the number of layers, H is the hidden size, and A is the number of attention heads.

**Comparison with Previous Work**    Next, we compare the effectiveness of our NAS-BERT to previous methods on BERT compression. Since they usually compress BERT into a model size of about

| Model | Params | MNLI | QQP | QNLI | CoLA | SST-2 | STS-B | RTE | MRPC | AVG |
|---|---|---|---|---|---|---|---|---|---|---|
| *dev set* | | | | | | | | | | |
| DistilBERT | 66M | 82.2 | 88.5 | 89.2 | 51.3 | 91.3 | 86.9 | 59.9 | 87.5 | 79.6 |
| MiniLM | 66M | 84.0 | 91.0 | 91.0 | 49.2 | 92.0 | - | 71.5 | 88.4 | - |
| BERT-of-Theseus | 66M | 82.3 | 89.6 | 89.5 | 51.1 | 91.5 | 88.7 | 68.2 | - | - |
| PD-BERT | 66M | 82.5 | 90.7 | 89.4 | - | 91.1 | - | 66.7 | 84.9 | - |
| DynaBERT* | 60M | 84.2 | **91.2** | 91.5 | 56.8 | 92.7 | 89.2 | 72.2 | 84.1 | 82.7 |
| NAS-BERT | 60M | 84.1 | 91.0 | 91.3 | 58.1 | 92.1 | 89.4 | 79.2 | 88.5 | 84.2 |
| NAS-BERT* | 60M | **84.8** | **91.2** | **91.9** | **58.7** | **93.1** | **89.9** | **79.8** | **88.9** | **84.8** |
| *test set* | | | | | | | | | | |
| BERT-of-Theseus | 66M | 82.4 | **89.3** | 89.6 | 47.8 | 92.2 | 84.1 | 66.2 | 83.2 | 79.4 |
| PD-BERT | 66M | 82.8 | 88.5 | 88.9 | - | 91.8 | - | 65.3 | 81.7 | - |
| BERT-PKD | 66M | 81.5 | 88.9 | 89.0 | - | 92.0 | - | 65.5 | 79.9 | - |
| TinyBERT* | 66M | **84.6** | 89.1 | 90.4 | **51.1** | **93.1** | 83.7 | 70.0 | 82.6 | 80.6 |
| NAS-BERT | 60M | 83.5 | 88.9 | 90.9 | 48.4 | 92.9 | 86.1 | **73.7** | 84.5 | 81.1 |
| NAS-BERT* | 60M | 84.1 | 88.8 | **91.2** | 50.5 | 92.6 | **86.9** | 72.7 | **86.4** | **81.7** |

Table 3: Results on the dev and test set of the GLUE benchmark. "*" means using data augmentation. The test set results are obtained from the official GLUE leaderboard.

| Setting | MNLI | QQP | QNLI | CoLA | SST-2 | STS-B | RTE | MRPC | AVG |
|---|---|---|---|---|---|---|---|---|---|
| w/o PS | 83.5 | 90.6 | 90.8 | 55.5 | 92.0 | 88.1 | 77.4 | 86.5 | 83.1 |
| w PS | 84.1 | 91.0 | 91.3 | 58.1 | 92.1 | 89.4 | 79.2 | 88.5 | **84.2** |

Table 4: The results of NAS-BERT with and without progressive shrinking (PS) on NAS-BERT$_{60}$.

66M or 60M, we use our NAS-BERT$_{60}$ for comparison. We mainly compare our NAS-BERT with 1) DistilBERT (Sanh et al., 2019), which uses knowledge distillation on the pre-training stage; 2) BERT-PKD (Sun et al., 2019), which distills the knowledge from the intermediate layers and the final output logits on the pre-training stage; 3) BERT-of-Theseus (Xu et al., 2020), which uses module replacement for compression; 4) MiniLM (Wang et al., 2020b), which transfers the knowledge from the self-attention module; 5) PD-BERT (Turc et al., 2019), which distills the knowledge from the target domain in BERT training; 6) DynaBERT (Hou et al., 2020), which uses network rewiring to adjust width and depth of BERT for each downstream task. 7) TinyBERT (Jiao et al., 2019), which leverage embedding layer, hidden layer, and attention matrix distillation to mimic the teacher model at both the pre-training and fine-tuning stages. To make comparison with DynaBERT and TinyBERT, we also use their data augmentation (Hou et al., 2020; Jiao et al., 2019) on downstream tasks. Table 3 reports the results on the dev and test set of the GLUE benchmark. Without data augmentation, NAS-BERT achieves better results on nearly all the tasks compared to previous work. Further, with data augmentation, NAS-BERT outperforms DynaBERT and TinyBERT. Different from these methods that leverage advanced knowledge distillation techniques in pre-training and/or fine-tuning, NAS-BERT mainly takes advantage of architectures and achieves better accuracy, which demonstrates the advantages of NAS-BERT in model compression.

### 4.3 ABLATION STUDY

| Setting | MNLI | QQP | QNLI | CoLA | SST-2 | STS-B | RTE | MRPC | AVG |
|---|---|---|---|---|---|---|---|---|---|
| PS-arch | 84.1 | 91.0 | 91.3 | 58.1 | 92.1 | 89.4 | 79.2 | 88.5 | **84.2** |
| PS-op | 83.8 | 91.0 | 90.2 | 54.5 | 92.0 | 88.8 | 75.6 | 86.1 | 82.8 |

Table 5: The results of different progressive shrinking approaches on NAS-BERT$_{60}$. PS-arch and PS-op denote pruning architectures and operations in progressive shrinking.

**Ablation Study on Progressive Shrinking** To verify the effectiveness of progressive shrinking (PS), we train the supernet with the same number of training epochs but without progressive

shrinking. We follow the same procedure in NAS-BERT for model selection and final evaluation on downstream tasks. Due to the huge evaluation cost during the model selection on the whole search space without progressive shrinking, it costs 50 hours (evaluation cost on the shrinked search space only takes 5 minutes since there are only 10 architectures remaining in each bin in each block). The results are shown in Table 4. It can be seen that NAS-BERT with progressive shrinking searches better architectures, with less total search time.

We further show the training loss curve in Fig. 3. It can be seen that the superset without progressive shrinking suffers from slow convergence. The huge number of architectures in the supernet need long time for sufficient training. Given a fixed budget of training time, progressive shrinking can ensure the promising architectures to be trained with more resources and result in more accurate evaluation, and thus better architectures can be selected. On the contrary, without progressive shrinking, the amortized training time of each architecture is insufficient, resulting in inaccurate evaluation and model selection.

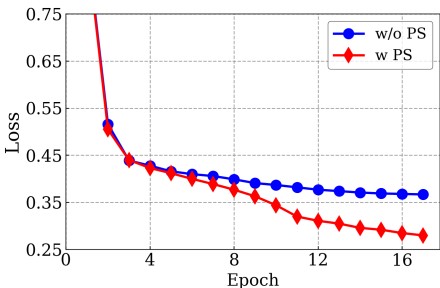

Figure 3: Loss curve of supernet training. Progressive shrinking starts at the epoch 4.

**Different Progressive Shrinking Approaches**   Instead of pruning architectures (paths) from the search space, we can also prune operations (nodes) from the supernet (Radosavovic et al., 2020; Luo et al., 2020) in progressive shrinking. From the perspective of supernet, the former is to remove paths and the latter is to remove nodes from the supernet. To evaluate the performance of operations (nodes) in supernet, at the end of each training epoch, we evaluate the architectures on the dev set and prune the search space according to the performance (validation loss) of operations. The validation loss of the operation $o_i$ in $l$-layer is estimated by the mean validation losses of all the architectures whose operation in the $l$-th layer $a^l = o_i$. The shrinking algorithm proceeds as follows:

- Sample $E$ architectures and get the validation losses on the dev set.

- Rank operations according to their mean validation losses in descending order.

- Prune operations with the largest losses from the supernet repeatedly until removing $R$ ($R$ is a hyper-parameter to control the speed of pruning) of the architectures in the search space.

The shrinking algorithm performs at the end of each training epoch, and terminates when only $m = 10$ architectures are left in each bin in each block, and the training also ends. For the fair comparison, we set $m = 10$ and $E = 1000$, which are same as settings in Section 4.1. At the end of each epoch, we perform this algorithm to remove $R = 30\%$ architectures for each bin. In this way, the algorithm terminates at the same epoch as that in Section 3.2.2. At the end of each training epoch, we evaluate the architectures on the dev set and prune the search space according to the performance of *operations*. As shown in Table 5, pruning architectures in progressive shrinking achieves better results.

## 5  CONCLUSION

In this paper, we propose NAS-BERT, which leverages neural architecture search (NAS) to compress BERT models. We carefully design a search space with different operations associated with different hidden sizes, to explore the potential of diverse architectures and derive models with adaptive sizes according to the memory and latency requirements of different devices. The compression is conducted on the pre-training stage and is downstream task agnostic, where the compressed models can be applicable for different downstream tasks. Experiments on the GLUE benchmark datasets demonstrate the effectiveness of our proposed NAS-BERT compared with both hand-designed BERT baselines and previous works on BERT compression. For future work, we will explore more advanced search space and NAS methods to achieve better performance.

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

## A  APPENDIX

### A.1  OPERATION SET AND SEARCH SPACE

**The Design Choice of the Operation Set**   In addition to MHA and FFN, LSTM, convolution and variants of MHA and FFN have achieved good performance in many NLP tasks (Chen et al., 2020; Kaiser et al., 2018; Karatzoglou et al., 2020; Bahdanau et al., 2014). We describe the considerations to choose the operations in Table 1 as follows: 1) LSTM is not considered due to slow training and inference speed. 2) In our preliminary experiments, we try some variants of MHA and FFN (Lample et al., 2019; Wu et al., 2018), but fail to observe the advantages of small model size and/or better performance. 3) Considering the parameter size of convolution is $K * H^2$ and separable convolution (SepConv) is $H^2 + K * H$ where $K$ and $H$ are the kernel size and hidden size, instead of convolution, we use SepConv with a larger kernel (with a larger receptive field) without significantly increasing the model size and the latency. Based on these considerations, we add SepConv into the candidate operation set.

To determine the possible hidden sizes for operations, we mainly consider the range of the compressed model sizes. Previous works (Sanh et al., 2019; Sun et al., 2019; Song et al., 2020; Jiao et al., 2019; Lan et al., 2019; Zafrir et al., 2019; Shen et al., 2020; Lan et al., 2019; Zafrir et al., 2019; Chen et al., 2020) usually compress pre-trained BERT model into a small model (usually no more than 66M) for efficiency and effectiveness. Similarly, in this work, we aim to obtain compressed models less than 66M. Therefore, we choose the hidden sizes between 128 and 512 for the candidate operations, which enables a good trade-off between efficiency and effectiveness[1].

**The Complexity of the Search Space**   The supernet consists of $L = 24$ layers. If we do not use block-wise search, there would be $26^{24} \approx 10^{34}$ paths (possible candidate architectures) in the supernet. We divide the supernet into $N = 4$ blocks and each block contains 6 layers. Within each block, the hidden size of the 6 layers are required to be consistent, and the hidden sizes across different blocks can be different. So there are $5 * 6^6 = 233280$ paths (possible candidate sub-architectures) in each block, where the first 5 is the number of candidate hidden sizes and $6^6$ represents that there are 6 operations in 6 layers. Due to the identity operation, there is an unnecessary increase in the possible sequence of operations (architecture) as pointed in Li et al. (2020a). For example, the architecture {FFN, identity, FFN, identity} is equivalent to {FFN, FFN, identity, identity}. Thus we only keep the architectures that all of the identity operations are at the end (e.g., {FFN, FFN, identity, identity}) and delete other redundant architectures. After cleaning the redundancy, the search space in each block is reduced from the original 233280 to 97650, which largely improves the sample efficiency. We can select sub-architectures from each block and ensemble them to get a complete model. Considering $N = 4$ blocks, there are $97650^4$ (about $10^{20}$ possible combinations). Therefore the number of possible models is reduced from $10^{34}$ to $10^{20}$.

### A.2  TRAINING CONFIGURATIONS

**Teacher Model**   We train the BERT$_{base}$ (L=12, H=768, A=12) (Devlin et al., 2019) as the teacher model, where L is the number of layers, H is the hidden size, and A is the number of attention heads. Following Devlin et al. (2019), we use BookCorpus plus English Wikipedia as pre-training data (16GB in total). We use Adam (Kingma & Ba, 2014) with a learning rate of 1e-4, $\beta_1 = 0.9$ and $\beta_2 = 0.999$. The learning rate is warmed up to a peak value of 5e-4 for the first 10,000 steps, and then linearly decays. The weight decay is 0.01 and the dropout rate is 0.1. We apply the best practices proposed in Liu et al. (2019) to train the BERT$_{base}$ on 16 NVIDIA V100 GPUs with large batches leveraging gradient accumulation (2048 samples per batch) and 125000 training steps. We present the performance of the teacher model and compare it with teacher models used in other works in Table 6. Our teacher model is better than others, which is mainly caused by the volatility of RTE and CoLA (small dataset). After removing these two datasets, the performance of the teacher model (average score: 89.74) is close to the teacher model of other methods (DistilBERT: 89.73, BERT-of-Theseus: 88.76 and DynaBERT: 89.68). In this way, NAS-BERT can still show its effectiveness compared with other approaches, without considering RTE and CoLA in Table 3.

---

[1]We do not use hidden size smaller than 128 since it cannot yield model with enough accuracy.

| Teacher model | MNLI-m | QQP | QNLI | CoLA | SST-2 | STS-B | RTE | MRPC | AVG |
|---|---|---|---|---|---|---|---|---|---|
| DistilBERT | 86.7 | 89.6 | 91.8 | 56.3 | 92.7 | 89.0 | 69.3 | 88.6 | 83.0 |
| MiNiLM | 84.5 | 91.3 | 91.7 | 58.9 | 93.2 | - | 68.6 | 87.3 | - |
| BERT-of-Theseus | 83.5 | 89.8 | 91.2 | 54.3 | 91.5 | 88.9 | 71.1 | 89.5 | 82.3 |
| DynaBERT | 84.8 | 90.9 | 92.0 | 58.1 | 92.9 | 89.8 | 71.1 | 87.7 | 83.4 |
| MobileBERT | 87.0 | - | 93.2 | - | 94.1 | - | - | 87.3 | - |
| Ours | 85.2 | 91.0 | 91.3 | 61.0 | 92.9 | 90.3 | 76.0 | 87.7 | 84.4 |

Table 6: The accuracy of the teacher models on dev set of the GLUE benchmark. The teacher model of MobileBERT is IB-BERT$_{LARGE}$ which reaches the similar accuracy as original BERT$_{LARGE}$.

### A.3 INFERENCE AND FLOPS SETUP

Following Sun et al. (2019); Wang et al. (2020b), the inference time is evaluated on QNLI training set with the batch size of 128 and the maximum sequence length of 128. The numbers reported in Table 2 are the average of 100 batches on Intel(R) Xeon(R) CPU E5-2690 v4 @ 2.60GHz with 12 cores. Following Clark et al. (2019), the inference FLOPs are calcuated with single length-128 input.

### A.4 TWO-STAGE DISTILLATION

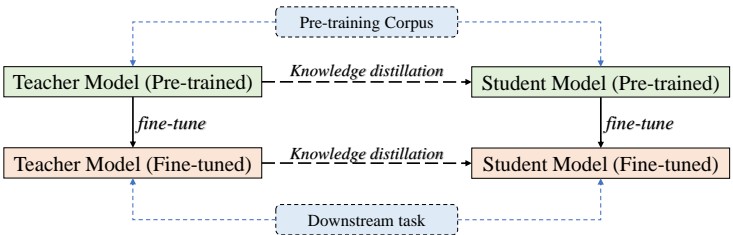

Figure 4: The pipeline of two-stage distillation.

Two-stage distillation means applying knowledge distillation in both the pre-training and the fine-tuning stage. Previous methods (Song et al., 2020; Sanh et al., 2019; Jiao et al., 2019; Gordon et al., 2020) have proved that using two-stage distillation is superior to the single-stage distillation. The pipeline of two-stage distillation is shown in Fig. 4. The procedure of two-stage distillation can be summarized as follows:

1. Pre-train the teacher model on the pre-training corpus.
2. Pre-train the light-weight student model with knowledge distillation from the pre-trained teacher model in step 1.
3. Fine-tune the pre-trained teacher model in step 1 on the downstream task.
4. Fine-tune the pre-trained student model in step 2 with knowledge distillation from the fine-tuned teacher model in step 3 on the target downstream task.

To simplify our expression, we denote the parameter of the student and the teacher model as $\theta_S$ and $\theta_T$ respectively. We adopt a general formulation for knowledge distillation in both stages:

$$\mathcal{L}(x, y; \theta_S, \theta_T) = \sum_{(x,y)}^{\{\mathcal{X}, \mathcal{Y}\}} (1 - \lambda) \cdot \mathcal{L}_{MLE}(x, y; \theta_S) + \lambda \cdot \mathcal{L}_{KL}(f(x; \theta_T), f(x; \theta_S)), \quad (2)$$

where $\mathcal{L}_{MLE}$ is the maximal likelihood loss of the student model $\theta_S$ over the training corpus $(\mathcal{X}, \mathcal{Y})$ (i.e., masked language model loss on the pre-training stage or classification/regression loss on the fine-tuning stage), and $\mathcal{L}_{KL}$ is the KL divergence between the predicted probability distribution $f(x; \theta_T)$ of the teacher model $\theta_T$ and the distribution $f(x; \theta_S)$ of the student model $\theta_S$. $\lambda$ is a hyper-parameter to trade off $\mathcal{L}_{MLE}$ and $\mathcal{L}_{KL}$, and is set as 0.5 in our experiments. There are other advanced distillation techniques (e.g., embedding distillation or attention distillation) but here we only consider prediction layer distillation.

A.5    OTHER DESIGN CHOICES IN NAS-BERT

**Why dividing the model into $N(= 4)$ blocks?**   If $N$ is small, the search space for each block will be huge (e.g., $C^L$ possible architectures when $N = 1$) and the training is costly. If $N$ is large, candidate architectures in each block will be very limited (e.g., $C$ architectures when $N = L$) and cannot explore the potential of combinations of different operations. We set $N = 4$ for a trade-off following Li et al. (2020a).

**Why dividing the search space in each block into $B(= 10)$ bins?**   If $B$ is too large, there are at least $m * B$ architectures during the whole process of supernet training. The amortized training time of each architecture can be insufficient, resulting in inaccurate evaluation and model selection as shown in Section 4.3. If $B$ is very small, we cannot get various architectures at the end of the training. Consequently, we set $B = 10$ for the trade-off.

**Why conducting progressive shrinking until $m(= 10)$ architectures is reached?**   As introduced in Section 3.3, we have $(m * B)^N$ possible combinations to build the lookup table $LT_{arch}$. When $m = 10$, the table has $10^8$ architectures for the selection, which is enough to select models under diverse requirements. If $m$ is too large, storing the big table $LT_{arch}$ with so many items $((m * B)^N)$ is costly and the evaluation cost in Section 3.3 will increase exponentially.

**Is the hidden size transformation module in Fig. 2 retained in the final model?**   The answer is No. In the final derived architecture, we re-train the architecture by two-stage distillation without adding a hidden size transformation module because the architecture does not need to match the hidden size to the one in the teacher model. However, there may be hidden size mismatch in the architecture itself due to the design of blocks, and we simply add an additional linear layer to handle it.

**How to decide the embedding layer on the model selection stage?**   In order to save computing resources, we do not search the width of the embedding layer in NAS-BERT. We manually design the embedding layer for the final derived architecture according to the requirement of compressed model size: 1) <10M, $W_E$=64, 2) <20M, $W_E$=128, 3) <35M, $W_E$=256, 4) <50M, $W_E$=384 and 5) >50M, $W_E$=512, where $W_E$ is the width of the embedding layer. For example, we choose the embedding layer with hidden size 512 when that target model size is required to be larger than 50M.

A.6    SEARCHED ARCHITECTURES BY NAS-BERT

Our NAS-BERT can generate different compressed models given specific constraints, as described in Section 3.3. Besides the several NAS-BERT models we evaluate in Table 2, we further select architectures with various sizes, from 5M to 60M with 5M interval, yielding totally 12 different architectures with different sizes. We present the architectures in the below figures (Figure (a)~(l)).

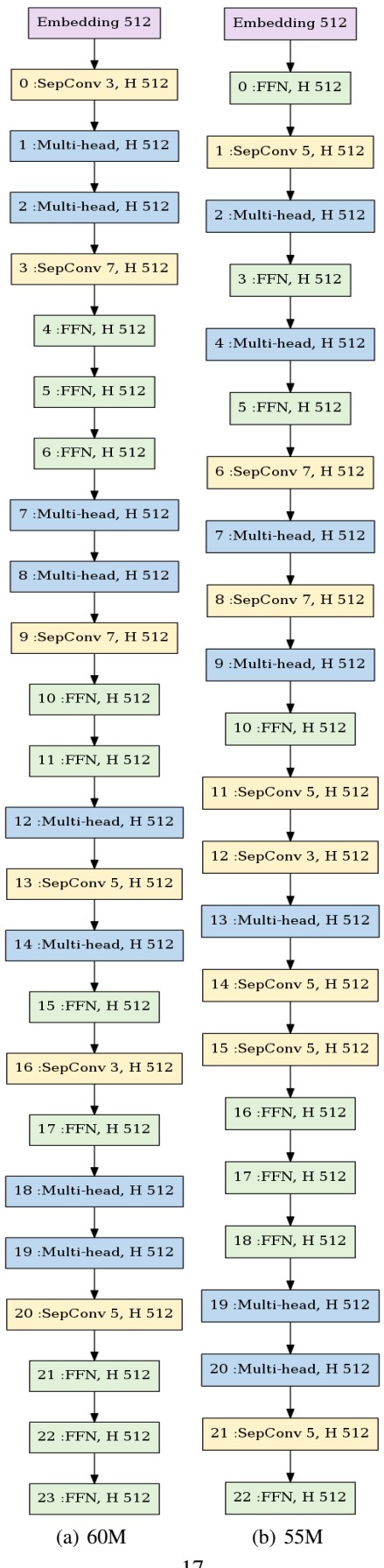

(a) 60M              (b) 55M

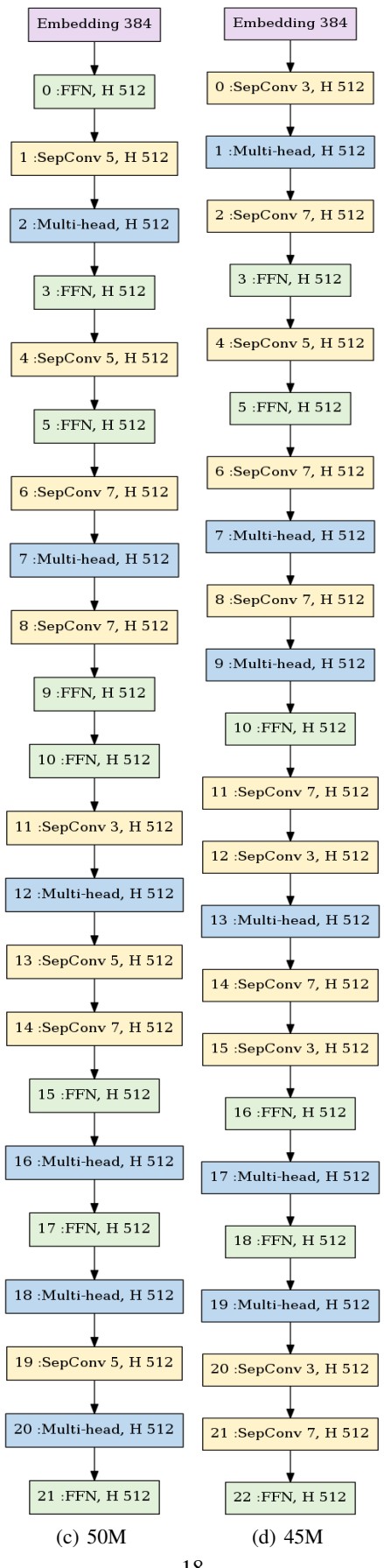

(c) 50M        (d) 45M

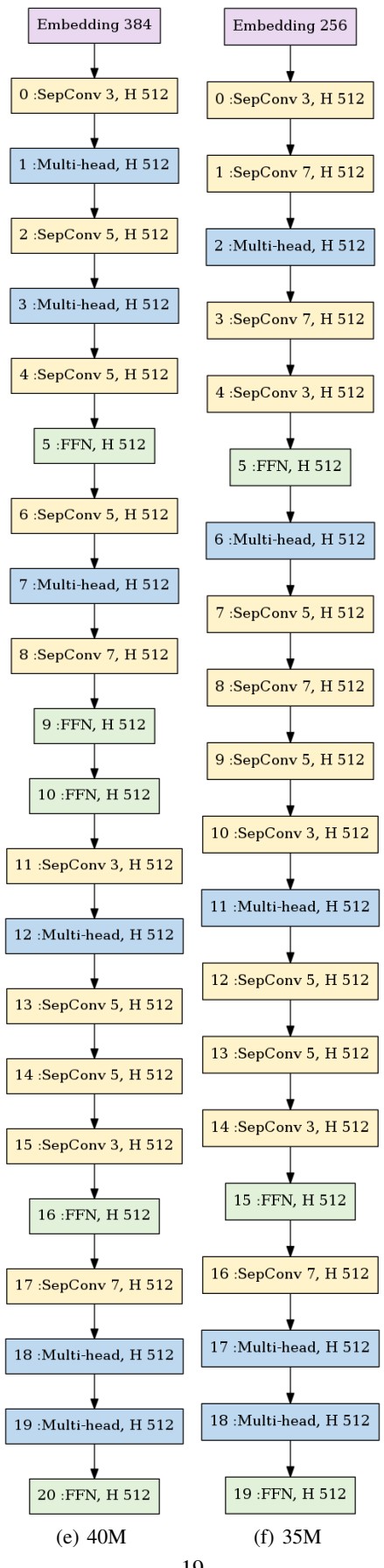

(e) 40M          (f) 35M

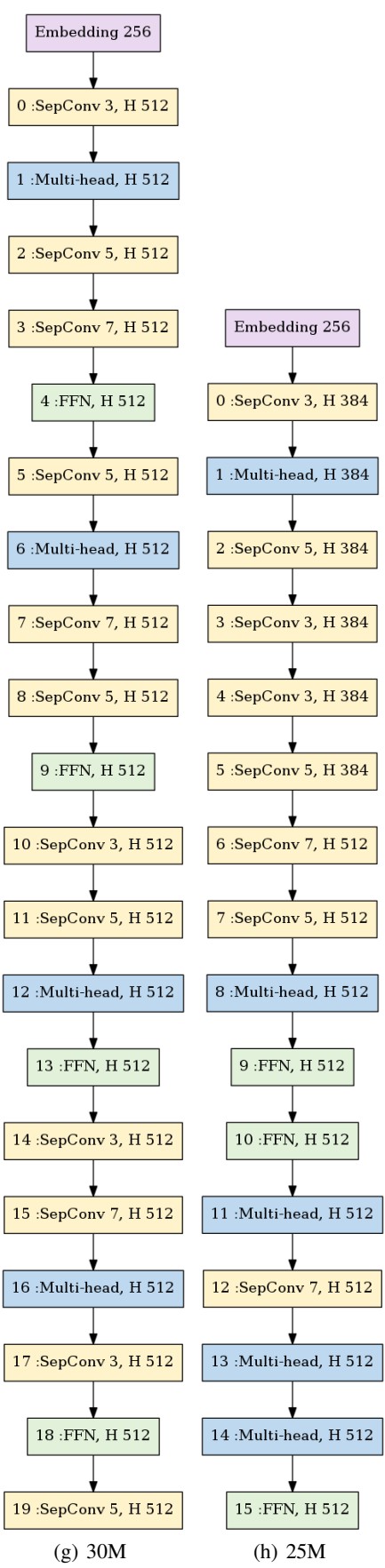

(g) 30M  (h) 25M

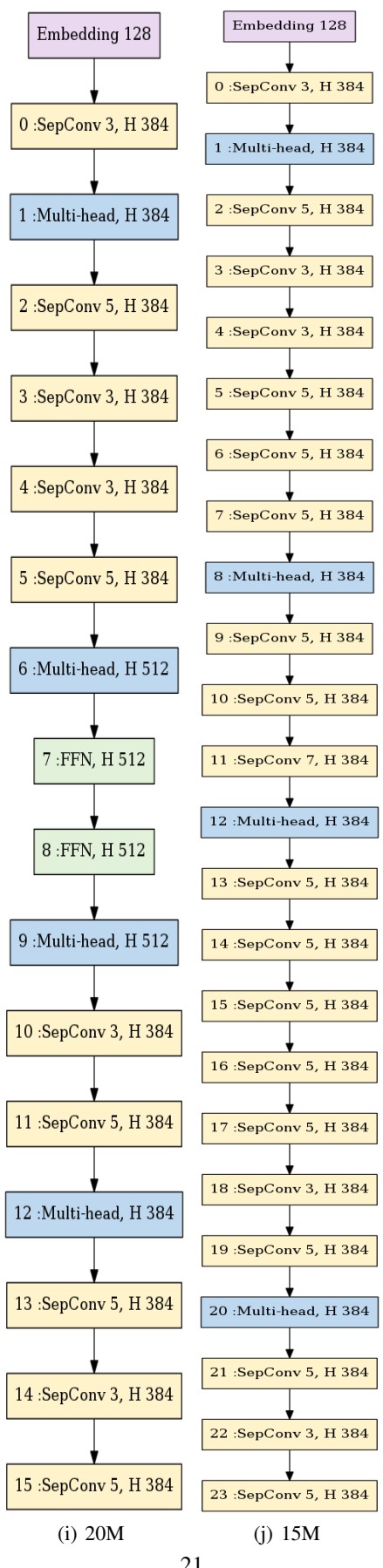

(i) 20M          (j) 15M

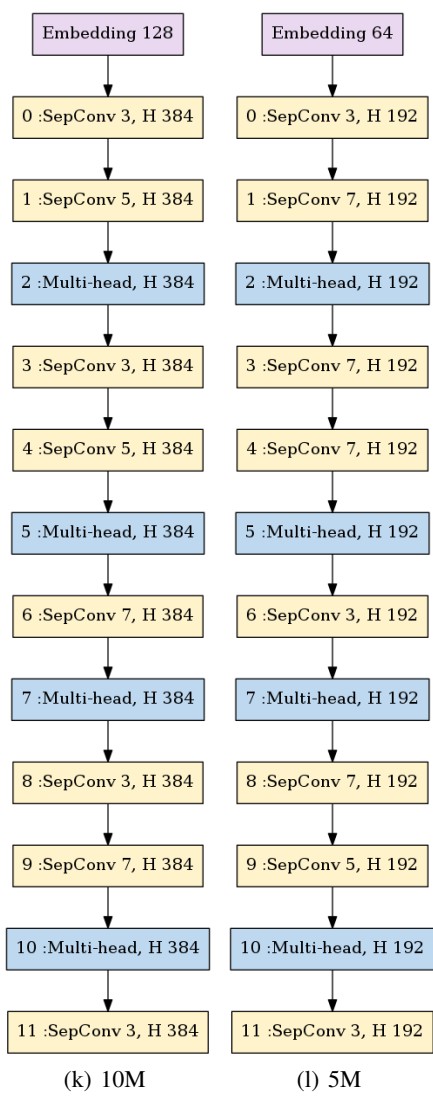

(k) 10M          (l) 5M

