# OpenReview forum: "Task-Agnostic and Adaptive-Size BERT Compression"
_ICLR.cc/2021/Conference — Reject_

### Official Review · AnonReviewer1 · 2020-10-27
**Paper on applying NAS to pre-training tasks**

**Rating:** 6
**Confidence:** 4

**Review:**

This paper presents an effective NAS method for pre-trained language models at the pre-training stage, so the selected models can be applied to various downstream tasks with fine-tuning. To achieve better performances the widely used two-stage distillation and data augmentation are applied to the selected models from super-net.

The main contribution of this work lies in the designed search space and the proposed three strategies (block-wise search, progressive shrinking and performance approximation) for improving search efficiency and accuracy. Although the novelty of this work is quite limited, training a big supernet for BERT at the pre-training stage is not trivial, which is useful for industry applications. The authors evaluate their approach on GLUE datasets and compare it to other state-of-the-art models.

The paper is well-written and organized, the experiments are thorough. However, I have several concerns:

1)  In the Table 1 under the KD setting, “two-stage distillation” is conducted on the selected models from supernet to further improve the performances, it would be interesting to add another two settings: a) only conducting the distillation at the pre-training stage, b) continuing to pre-training on large scale unlabeled data to finally obtain better task-agnostic models.

2) The models are evaluated on the GLUE dataset, more experiments on challenging QA tasks should be added.

3) In the Table 1, the comparison to MobileBERT and TinyBERT should be added, and the FLOPs or the inference time on CPU/GPU can be provided.

4) Some important related work should be included, [1] HAT: Hardware-Aware Transformers for Efficient Natural Language Processing (although this work focuses on the machine translation task) [2] Finding Fast Transformers: One-Shot Neural Architecture Search by Component Composition.

---

> ### Author Response · Authors · 2020-11-20
> **Response to AnonReviewer1 (Part 1)**
>
> Thanks for your constructive review comments. Below are our responses to your concerns.
>
> **[Ablation study of two-stage distillation]**
>
> To study the performance of distillation at the pre-training or fine-tuning stage, we conduct experiments on two settings: 1) only using the distillation on the pre-training stage; 2) only using the distillation on the fine-tuning stage. The results are shown as follows:
>
> | Setting    |\|| PD |\|| KD |\|| MNLI |\||  QQP |\|| QNLI |\|| CoLA |\|| SST-2 |\|| STS-B |\||  RTE |\|| MRPC |\||  AVG |
> |------------|:-:|:--:|:-:|:--:|:-:|:----:|:-:|:----:|:-:|:----:|:-:|:----:|:-:|:-----:|:-:|:-----:|:-:|:----:|:-:|:----:|:-:|:----:|
> | $\text{BERT}_{60}$     |\||  √ |\||  √ |\|| 83.2 |\|| 90.5 |\|| 90.2 |\|| 56.3 |\||  91.8 |\||  88.8 |\|| 78.5 |\|| 88.5 |\|| 83.5 |
> | $\text{NAS-BERT}_{60}$ |\||  √ |\||  √ |\|| 84.1 |\|| 91.0 |\|| 91.3 |\|| 58.1 |\||  92.1 |\||  89.4 |\|| 79.2 |\|| 88.5 |\|| **84.2** |
> | $\text{BERT}_{60}$     |\||  √ |\||    |\|| 83.2 |\|| 90.3 |\|| 89.5 |\|| 55.0 |\||  91.6 |\||  88.6 |\|| 77.8 |\|| 87.3 |\|| 82.9 |
> | $\text{NAS-BERT}_{60}$ |\||  √ |\||    |\|| 83.3 |\|| 90.9 |\|| 91.3 |\|| 55.6 |\||  92.0 |\||  88.6 |\|| 78.5 |\|| 87.5 |\|| **83.5** |
> | $\text{BERT}_{60}$     |\||    |\||  √ |\|| 83.1 |\|| 90.4 |\|| 90.4 |\|| 54.3 |\||  91.2 |\||  88.7 |\|| 75.6 |\|| 87.0 |\|| 82.6 |
> | $\text{NAS-BERT}_{60}$ |\||    |\||  √ |\|| 83.7 |\|| 90.8 |\|| 91.0 |\|| 54.2 |\||  92.1 |\||  89.4 |\|| 76.0 |\|| 87.5 |\|| **83.1** |
>
> **Title**: “PD” or “FD” means to add knowledge distillation on the pre-training or the fine-tuning stage.
>
> We have two observations from the results: 1) NAS-BERT outperforms the BERT baseline at different settings; 2) distillation on either pre-training or fine-tuning stage can improve the scores, and two-stage distillation can further get better performance.
>
> **[Continuing to pre-training on large scale unlabeled data]**
>
> We are now continuing to pre-train on the large-scale dataset (160GB)  to obtain a better model. Due to the huge training cost, we will update the results when it is finished.
>
> **[More results on the challenging QA tasks]**
>
> We follow your suggestions and are conducting experiments on SQuAD 1.1 and SQuAD 2.0. We will update the results when it is finished.

---

> > ### Author Response · Authors · 2020-11-20
> > **Response to AnonReviewer1 (Part 2)**
> >
> > **[Comparison with MobileBERT and TinyBERT; Adding FLOPs and Latency in Table 1]**
> >
> > **TinyBERT**
> >
> > Note that TinyBERT used sophisticated distillation techniques such as hidden layer distillation, attention score distillation. Even so, our NAS-BERT (60M) outperforms TinyBERT (66M) on GLUE tasks in terms of the average score as follows. We also update the result in the revised paper.
> >
> > | Model    |\|| Params |\|| MNLI |\||  QQP |\|| QNLI |\|| CoLA |\|| SST-2 |\|| STS-B |\||  RTE |\|| MRPC |\||  AVG |
> > |----------|:-:|:------:|:-:|:----:|:-:|:----:|:-:|:----:|:-:|:----:|:-:|:-----:|:-:|:-----:|:-:|:----:|:-:|:----:|:-:|:----:|
> > | TinyBERT |\||   66M  |\|| 84.6 |\|| 89.1 |\|| 90.4 |\|| 51.1 |\||  93.1 |\||  83.7 |\|| 70.0 |\|| 82.6 |\|| 80.6 |
> > | NAS-BERT |\||   60M  |\|| 84.1 |\|| 88.8 |\|| 91.2 |\|| 50.5 |\||  92.6 |\||  86.9 |\|| 72.7 |\|| 86.4 |\|| **81.7** |
> >
> > **MobileBERT**
> >
> > For MobileBERT, it is unfair to compare with it because the teacher model (IB-BERT) of MobileBERT achieves the performance close to $\text{BERT}_{\text{large}}$, which is much better than our teacher model and those used in most related works such as TinyBERT, DynaBERT, DistilBERT. We will also use a teacher model with similar performance for our NAS-BERT and compare it with MobileBERT, which we leave for future work.
> >
> > **FLOPs and Latency**
> >
> > We have added the latency and FLOPs in Table 2  in the revised version (i.e., Table 1 in the submitted version).
> >
> > **[More discussion with related work]**
> >
> > Thanks for our suggestions! We have cited the two papers [1,2] and made discussions in our updated version.
> >
> >
> >
> > ---------Work in Progress---------
> >
> > -Pre-train on the large-scale dataset (160GB)  to obtain better model.
> >
> > -Experiments on the SQuAD 1.1 and SQuAD 2.0
> >
> > [1] HAT: Hardware-Aware Transformers for Efficient Natural Language Processing
> > [2] Finding Fast Transformers: One-Shot Neural Architecture Search by Component Composition and provided some discussion between these methods.

---

> > > ### Comment · AnonReviewer1 · 2020-11-24
> > > **My concerns have been addressed**
> > >
> > > Thanks for your reply. Your explanation well addressed my concerns, so I have updated my score to 6.

---

> > > > ### Author Response · Authors · 2020-11-24
> > > > **The results of SQuAD v1.0 and v2.0**
> > > >
> > > > We get results of SQuAD 1.1 and SQuAD 2.0 are as follows:
> > > >
> > > > |Model|\|| Params   |\| | SQuAD v1.1| \|| SQuAD v2.0  |
> > > > |:----|:-:|:---: |:--:|:----:|:----: |:---: |
> > > > |========= |\||=======|\||==EM/F1== |\||==EM/ F1==|
> > > > |Teacher     | \||110M|\||81.8/88.9|\||74.5/77.9|
> > > > |DistilBERT |\||66M|\||79.1/86.9|\||-/-|
> > > > |BERT-PKD |\||66M|\||77.1/85.3|\||66.3/69.8|
> > > > |MiniLM†    |\||66M|\||-/-|\||-/76.4  |
> > > > |TinyBERT   |\||66M|\||79.7/87.5|\||69.9/73.4|
> > > > |NAS-BERT  |\||60M|\||80.5/88.0|\||73.2/76.3|
> > > > |NAS-BERT†|\||60M|\||**81.2**/**88.4**|\||**73.9**/**77.1**|
> > > >
> > > > **Title**: MiniLM† means that MiniLM in the original paper [1] was trained with more computations (batch size 1024 * 400,000 steps). Thus, we also trained the NAS-BERT† with the same computations (batch size 2048 * 200,000 steps) for a fair comparison.
> > > >
> > > >  We can find that our NAS-BERT, without using sophisticated distillation techniques, outperforms previous works with even slightly more parameters on both SQuAD v1.1 and v2.0.
> > > >
> > > > [1] Wang W, Wei F, Dong L, et al. Minilm: Deep self-attention distillation for task-agnostic compression of pre-trained transformers[J]. arXiv preprint arXiv:2002.10957, 2020.

---

### Official Review · AnonReviewer4 · 2020-10-28
**Interesting work on NAS for BERT**

**Rating:** 7
**Confidence:** 3

**Review:**

Summary:
This paper proposes to search architectures of BERT model under various memory and latency contraints. The search algorithm is conducted by pretraining a big supernet that contains the all the sub-network structures, where the optimal models for different requirements are selected from it. Once an architecture is found, it is re-trained through pretraining-finetuning or two-stage distillation for each specific task. Several approaches (block-wise training and search, progressive shrinking, performance approximation) are proposed to improve the search efficiency. Experiments on GLUE benchmark shows the models found by proposed methods can achieve better accuracy than some of the previous compressed BERT models. The paper (together with the appendix) is clearly presented, and the idea is new and interesting to me. The experiments are detailed and comprehensive.

Pros:
The paper is well presented. The architecture of the superent and the candidate operations are carefully designed and selected. It seems that the SpeConv operation is particularly effective when the model size is small. The search algorithm including the block-wise training, progressive shrinking can remove less-optimal structures quickly and significantly reduce the search space. The performance of NAS-BERT models are generally better than those of the compressed BERT models with similar model size, although the comparisons may not be completely fair.

Concerns:
1. The organization of the paper can be further improved. The paper may not be easy to follow if the appendix is skipped, especially for the readers who are not familiar with NAS or related work. Many of the important information can only be found in appendix.
2. The novelty of the paper is unclear to me. Although this work may be new on search BERT-like language model, it seems many of the ideas such as block-wise search and distillation are borrowed from existing work. Please the author clarify the main novelties and technical contribution of this work, especially to the field of neural architecture search or more broadly, AutoML . Moreover, some of the proposed techniques such as progressive shrinking are merely empirical practices and are lack of theory or insight showing how accurate the approximation would be.
3. It is usually more illustrative (and also space saving) to plot accuracy versus latency/#parameters of different models in the same figure. Some of the well noted models such as MobileBERT and TinyBERT are not included in comparison. For DynaBERT, there are multiple configurations but only one is included. AdaBERT, which adopts NAS for each specific task, should also be included if possible. Again, since there are of many models with different size and latency, it may be better to have a plot for clear comparison.
4. HAT (Wang et al. HAT: Hardware-Aware Transformers for Efficient Natural Language Processing. ACL 2020.) is not mentioned in the paper, which share similarities (training supernet) and differences (search algorithm) with this work from technical point of view. It will be better if the author can explain and compare the proposed search algorithm to evolutionary search.

---

> ### Author Response · Authors · 2020-11-20
> **Response to AnonReviewer4 (Part 1)**
>
> Thanks for your helpful review comments. Below are our responses to your concerns.
>
> **[Organization of the paper]**
>
> Thanks for your advice. We have re-organized in the new version as follows: 1) We move the detailed description of search space design from the Appendix to Section 3.1. 2) To enable better comparison of accuracy versus latency/#parameters of different models, we add more information in the Table 2 in the revised paper. 3) We move the experiment description of progreive shrinking from the Appendix to Section 4.2. 4) We add the discussions with other works including HAT in the related work part.
>
> **[About the novelty of NAS-BERT]**
>
> - To achieve task-agnostic and adaptive-size BERT compression, we conduct neural architecture search by training the big supernet on the pre-training stage, which is extremely costly. Our proposed techniques including block-wise search, progressive shrinking, and performance approximation all target for alleviating the pain of heavy training cost and improving the search efficiency.
>     - Different from block-wise search in [1], we add identity operation to automatically search models with adaptive depths in each block. [1] achieves this by manually training different depths to get several models, which is acceptable in lightweight image tasks, but is extremely costly in BERT pre-training task and cannot support devices with different kinds of memory and latency limitations.
>     - Besides block-wise search, our newly proposed progressive shrinking and performance approximation are critical for efficiency and effectiveness. Specifically, for progressive shrinking, we split the models into different bins to prevent the smaller models from being pruned and guarantee that we can get various architectures to meet different constraints.
> - We think our work is not simply using existing techniques, but developing the existing techniques (block-wise search) and designing new techniques (progressive shrinking, performance approximation, novel search space for BERT) to solve the unique challenges in task-agnostic and adaptive-size BERT compression.
> - From the field of neural architecture search, most algorithms [2,3,4] are proposed to improve the performance on the computer vision tasks and light-weight NLP tasks (e.g., PTB). We extend NAS to the much heavy BERT pre-training task and propose a series of techniques to improve the efficiency, which may inspire a lot of following works on NAS for heavy tasks.

---

> > ### Author Response · Authors · 2020-11-20
> > **Response to AnonReviewer4 (Part 2)**
> >
> > **[Comparison with DynaBERT, TinyBERT, MobileBERT and AdaBERT]**
> >
> > **DynaBERT**
> >
> > DynaBERT has multiple configurations (40M-110M) for each specific task and has an overlap (40M-60M) compared with NAS-BERT. Thus we use $\text{NAS-BERT}_{30,60}$ to compare with DynaBERT as follows:
> >
> > | Model     |\|| Params |\|| MNLI |\||  QQP |\|| QNLI |\|| CoLA |\|| SST-2 |\|| STS-B |\||  RTE |\|| MRPC |\||  AVG |
> > |-----------|:-:|:------:|:-:|:----:|:-:|:----:|:-:|:----:|:-:|:----:|:-:|:-----:|:-:|:-----:|:-:|:----:|:-:|:----:|:-:|:----:|
> > | DynaBERT* |\||   60M  |\|| 84.2 |\|| 91.2 |\|| 91.5 |\|| 56.8 |\||  92.7 |\||  89.2 |\|| 72.2 |\|| 84.1 |\|| 82.7 |
> > | NAS-BERT  |\||   60M  |\|| 84.1 |\||  91.0  |\|| 91.3 |\|| 58.1 |\||  92.1 |\||  89.4 |\|| 79.2 |\|| 88.5 |\|| 84.2 |
> > | NSA-BERT* |\||   60M  |\|| 84.8 |\|| 91.2 |\|| 91.9 |\|| 58.7 |\||  93.1 |\||  89.9 |\|| 79.8 |\|| 88.9 |\|| **84.8** |
> > | DynaBERT* |\||   40M  |\||  82.0  |\|| 90.4 |\|| 88.5 |\|| 43.7 |\||   92.0  |\||   87.0 |\|| 63.2 |\|| 81.4 |\|| 78.5 |
> > | NAS-BERT  |\||   30M  |\||  81.0  |\|| 90.2 |\|| 88.4 |\|| 48.7 |\||  90.5 |\||  87.6 |\|| 71.8 |\|| 84.6 |\|| 80.3 |
> > | NAS-BERT* |\||   30M  |\||  82.0  |\|| 90.4 |\||  89.0  |\|| 46.3 |\||  92.1 |\||  88.9 |\|| 73.5 |\|| 88.7 |\|| **81.4** |
> >
> > '*' means using data augmentation.
> >
> > We can find that our models achieve better performance than DynaBERT and especially, NAS-BERT with only 30M parameters outperforms DynaBERT with 40M parameters by 2.9 average GLUE score.
> >
> > **TinyBERT**
> >
> > Note that TinyBERT used sophisticated distillation techniques such as hidden layer distillation, attention score distillation. Even so, our NAS-BERT (60M) outperforms TinyBERT (66M) on GLUE tasks in terms of the average score as follows. We also update the results in the revised paper.
> >
> > | Model    |\|| Params |\|| MNLI |\||  QQP |\|| QNLI |\|| CoLA |\|| SST-2 |\|| STS-B |\||  RTE |\|| MRPC |\||  AVG |
> > |----------|:-:|:------:|:-:|:----:|:-:|:----:|:-:|:----:|:-:|:----:|:-:|:-----:|:-:|:-----:|:-:|:----:|:-:|:----:|:-:|:----:|
> > | TinyBERT |\||   66M  |\|| 84.6 |\|| 89.1 |\|| 90.4 |\|| 51.1 |\||  93.1 |\||  83.7 |\|| 70.0 |\|| 82.6 |\|| 80.6 |
> > | NAS-BERT |\||   60M  |\|| 84.1 |\|| 88.8 |\|| 91.2 |\|| 50.5 |\||  92.6 |\||  86.9 |\|| 72.7 |\|| 86.4 |\|| **81.7** |
> >
> > **MobileBERT**
> >
> > For MobileBERT, it is unfair to compare with it because the teacher model (IB-BERT) of MobileBERT achieves the performance close to the $\text{BERT}_{\text{large}}$, which is much better than our teacher model and those used in most related works such as TinyBERT, DynaBERT, DistilBERT. Anyway, we will also use a teacher model with similar performance for our NAS-BERT and compare with MobileBERT, which we leave for future work.
> >
> > **AdaBERT**
> >
> > For AdaBERT [5], it searches a task-specific architecture (6M - 10M) for each task and also introduces the special distillation techniques (e.g., using probe classifiers to hierarchically decompose the task-useful knowledge from the teacher model) and data augmentation. For fair comparison, we compare NAS-BERT (5M, without two-stage distillation) with AdaBERT (without data augmentation and probe classifiers, i.e., the results from Table 4 in [5]) as follows.
> >
> > | Setting  |\|| QNLI/Params |\|| MRPC/Params |\|| RTE/Params |
> > |----------|:-:|:-------------:|:-:|:-------------:|:-:|:------------:|
> > | AdaBERT  |\|| 82.0/7.9M   |\|| 77.2/7.5M   |\|| 56.7/8.6M  |
> > | NAS-BERT |\|| 83.9/5.0M   |\|| 80.0/5.0M   |\|| 67.0/5.0M  |
> >
> > We can find that NAS-BERT (5M) outperforms AdaBERT with even slightly more parameters. For future work, we will also compare NAS-BERT with AdaBERT using its special distillation techniques and data augmentation.

---

> > > ### Author Response · Authors · 2020-11-20
> > > **Response to AnonReviewer4 (Part 3)**
> > >
> > > **[Discussion with related work - HAT]**
> > >
> > > HAT [6] also trains a big supernet to generate various architectures. There are three differences between NAS-BERT and HAT:
> > > - From the perspective of tasks, HAT mainly focuses on the machine translation task but NAS-BERT focuses on the BERT pre-training task. More importantly, BERT pre-training is extremely heavy.  Therefore, we propose block-wise search, progressive shrinking and performance approximation to handle it.
> > > - From the perspective of the delivered architectures, HAT aims to generate models with different widths and depths, while NAS-BERT aims to generate novel architectures including width, depth, and novel combination of operations.
> > > - From the perspective of search algorithms, HAT uses an evolutionary search algorithm (a heuristic algorithm) with the help of a latency predictor, considering it is exhausting to search over all the models in the supernet. Different from HAT, NAS-BERT progressively prunes the search space during the supernet training stage and leaves more computation resources to the promising architectures, considering the huge cost of BERT pre-training task, and most architectures are not good enough and not necessary to be kept till the model selection stage. In the model selection stage, we only have limited but good architectures and can directly build a look-up table and select an architecture under given constraints.
> > > We also add the discussion and comparison with HAT [6] in our revised paper.
> > >
> > > [1] Li C, Peng J, Yuan L, et al. Block-wisely Supervised Neural Architecture Search with Knowledge Distillation[C]//Proceedings of the IEEE/CVF Conference on Computer Vision and Pattern Recognition. 2020: 1989-1998.
> > > [2] Pham H, Guan M Y, Zoph B, et al. Efficient Neural Architecture Search via Parameter Sharing[C]//ICML. 2018.
> > > [3] Cai H, Zhu L, Han S. ProxylessNAS: Direct Neural Architecture Search on Target Task and Hardware[C]//International Conference on Learning Representations. 2018.
> > > [4] Xu Y, Xie L, Zhang X, et al. Pc-darts: Partial channel connections for memory-efficient differentiable architecture search[J]. arXiv preprint arXiv:1907.05737, 2019.
> > > [5] Chen D, Li Y, Qiu M, et al. Adabert: Task-adaptive bert compression with differentiable neural architecture search[J]. arXiv preprint arXiv:2001.04246, 2020.
> > > [6] HAT: Hardware-Aware Transformers for Efficient Natural Language Processing

---

### Official Review · AnonReviewer2 · 2020-10-28

**Rating:** 6
**Confidence:** 5

**Review:**

Summary

The paper develops a new method to compress the BERT model with varying model sizes depending on the underlying usage. They use block-wise neural architecture search to choose the best set of submodules for each of the blocks. To reduce the size of the exponential search space they progressively remove the architectural configurations that yields high loss. Over-all the paper is well written and nicely presented.

+ve

- The NAS-BERT can produce pertained models with varying model sizes which is better than DistilBERT and BERT-PKD that requires pre-training every time the number of layers are varied.
- The paper conducts rigorous experiments to demonstrate the effectiveness of NAS-BERT. The baseline methods used for comparison covers most of the state-of-the-art methods used for model compression for BERT.

Concerns:

- Can the authors clarify if they save the model parameters corresponding to all the possible architectural choices? Or they find out the best configuration matching the model size and latency requirements and then do the pre-training again with those architectural choices.

- In appendix A.7 the paper demonstrates some of the architectures used by NAS-BERT. For lower model size ( < 20M ) it can be observed that the NAS-BERT ends up choosing SepConv layers most of the times. Do the authors do any analysis on why SepConv layer works better than the self-attention layer and the feed forward network. How does the network perform if it is composed of all SepConv layers? Have the authors tried to use only SepConv layers and see if that itself gives good accuracy rather than doing the architecture search.

- In terms of original ideas, although the concepts of block-wise architecture search, using SepConv layer for NLP tasks and using block-wise knowledge distillation are not novel by themselves but this paper has efficiently made use of the available techniques (along with efficient engineering work like progressively reducing the search space) to develop a method that gives good performance on NLP tasks.

---

> ### Author Response · Authors · 2020-11-20
> **Response to AnonReviewer2 (Part 1)**
>
> Thanks for your constructive review comments. Below are our responses to your concerns.
>
> **[clarify if we save the model parameters corresponding to all the possible architectural choices]**
>
> NAS-BERT can deliver various architectures with different model sizes and latency, which can save model parameters in different degrees. According to the given constraints (params, latency), we can select an architecture to meet these requirements. Then the selected model is pre-trained from scratch independently without inheriting the parameters from the supernet.
>
> **[Analysis on why SepConv is preferred in small models]**
>
> We have conducted the analysis before the search space design. To demonstrate why SepConv is preferred in small models, we show the parameters and latency of MHA, FFN and SepConv as follows:
>
> | Operation (hidden size 384) |\||  Params |\||  Latency (s) |\|| Latency/Params |
> |-----------------------------|:--:|:-------:|:--:|:--------:|:--:|:--------------:|
> | Multi-Head Attention        |\||  592128 |\|| 0.006146 |\||    1.00E-08    |
> | Feed-Forward Network        |\|| 1182336 |\|| 0.005555 |\||    4.70E-09    |
> | Separable Conv 3         |\||  298760 |\|| 0.002272 |\||    7.60E-09    |
> | Separable Conv 5            |\||  300300 |\|| 0.002285 |\||    7.60E-09    |
> | Separable Conv 7            |\||  301840 |\|| 0.002297 |\||    7.60E-09    |
>
> Compared with MHA, we can find that Latency/Params ratio of SepConv is smaller. Thus, the latency of SepConv is low when using the same number of parameters. Compared with FFN,  given constraints of the same number of parameters, we can stack more layers without adding much latency (usually better representation ability). For small models with strict latency and parameter constraints, SepConv can be stacked with more layers than FFN and MHA, and do not add too much latency. Thus SepConv is preferred. However, the network completely composed of SepConv layers cannot get good results as explained in the next point below.
>
> **[Can the network completely composed of SepConv layers get better results than the searched model?]**
>
> In our preliminary study, we build a network completely composed of SepConv layers, which cannot get good performance. Specifically, we build two 10M models with a full stack of SepConv layers: 1) SepConv with kernel size 3 and hidden size 384; 2) SepConv with kernel size 7 and hidden size 384 to compare with $\text{NAS-BERT}_{10}$. The results are as follows:
>
> | Setting            |\|| MNLI |\||  QQP |\|| QNLI |\|| CoLA |\|| SST-2 |\|| STS-B |\||  RTE |\|| MRPC |\||  AVG |
> |--------------------|:-:|:----:|:-:|:----:|:-:|:----:|:-:|:----:|:-:|:-----:|:-:|:-----:|:-:|:----:|:-:|:----:|:-:|:----:|
> | $\text{NAS-BERT}_{10}$         |\|| 76.4 |\|| 88.5 |\|| 86.3 |\||  34.0  |\||  88.6 |\||  84.8 |\|| 66.6 |\|| 79.1 |\|| **75.5** |
> | $\text{BERT}_{10}$             |\|| 74.4 |\|| 87.8 |\|| 85.7 |\|| 32.5 |\||  86.6 |\||  85.2 |\|| 66.9 |\|| 77.9 |\|| 74.6 |
> | SepConv (Kernel 3) |\|| 58.1 |\|| 80.9 |\|| 63.0 |\|| 38.4 |\||  83.3 |\||  45.2 |\|| 53.0 |\|| 70.7 |\|| 61.6 |
> | SepConv (Kernel 7) |\|| 64.7 |\|| 82.8 |\|| 67.8 |\|| 40.7 |\||  85.4 |\||  51.2 |\|| 54.7 |\|| 73.8 |\|| 65.1 |
>
> We can observe that SepConv network gets much worse performance on the paired sentence tasks (MNLI, QQP, QNLI, STS-B, RTE and MRPC) compared with NAS-BERT and BERT baseline. We analyze that the paired sentence tasks typically need a large reception field to efficiently extract the information between two sentences. However, network composed of only SepConv has limited reception field given normal kernel size and limited layers, which cannot efficiently model the global information. We can find that SepConv network with kernel size 7 gets better scores than that with kernel size 3 but still has a large gap with BERT baseline and NAS-BERT. We hypothesis that the advantages of NAS-BERT rely on the novel stacking between MHA, SepConv and FFN. MHA can extract the global information and SepConv or FFN can refine the information between adjacent positions (feed-forward network can be viewed as a convolutional network with kernel size 1).

---

> > ### Author Response · Authors · 2020-11-20
> > **Response to AnonReviewer2 (Part 2)**
> >
> > **[About the novelty of NAS-BERT]**
> >
> > - Applying neural architecture search directly on the heavy pre-training task is challenging. Our proposed techniques including block-wise search, progressive shrinking, and performance approximation all target for alleviating the pain of heavy training cost and improving the search efficiency.
> > - Different from block-wise search in [1], we add identity operation to automatically search models with adaptive depths in each block, which is not supported by [1], and thus [1] cannot automatically search models with adaptive depths. [1] solves this problem by manually training different depths to get several models, which is acceptable in lightweight image tasks, but is extremely costly in BERT pre-training task and cannot support devices with different kinds of memory and latency limitations.
> > - Note that besides block-wise search, our newly proposed progressive shrinking and performance approximation are critical for efficiency and effectiveness. Specifically, for progressive shrinking, we split the models into different bins to prevent the smaller models from being pruned and guarantee that we can get various architectures to meet different constraints.
> >
> > We think our work is not simply using existing techniques, but developing the existing techniques and designing new techniques to solve the unique challenges in task-agnostic and adaptive-size BERT compression.
> >
> > [1] Li C, Peng J, Yuan L, et al. Block-wisely Supervised Neural Architecture Search with Knowledge Distillation[C]//Proceedings of the IEEE/CVF Conference on Computer Vision and Pattern Recognition. 2020: 1989-1998.

---

### Official Review · AnonReviewer3 · 2020-10-29
**Reasonable techniques and decent performance but with quite some complication**

**Rating:** 5
**Confidence:** 5

**Review:**

First of all, I believe the paper is looking into a very important question that attracts lots of attention recently. The set of techniques proposed in the work are also reasonable and practical, where the proposed progressive space pruning seems to work very well. Empirically, the obtained models do perform better compared to standard Transformer baselines. As for the comparison with previous methods, since there are too many implementation details that can affect the fairness of the comparison (e.g. length of pretraining, batch size, teacher performances, etc), it's hard to judge the actual scale of the gain.

There are also a few concerns.
- Firstly, when the block-wise search is used, it feels like the NAS-BERT is trained in a way that is more similar to a variant of distillation that additionally utilizes intermediate hidden states. As this signal is not used in the standard BERT baseline, some improvement could actually come from this factor besides a better model (architecture+param). A better baseline could be a Transformer trained in a similar way.
- Secondly, in terms of novelty, this work is more like the combination of existing ideas, namely "once-for-all" and "block-wise search". One general issue with "once-for-all" is after the search, although you obtain multiple models of different sizes, the excellence these models are tied to the (1) specific set of shared parameters obtained and (2) the pre-training task. So, whether this is really a good way to obtain the desired task-agnostic compressed models is still questionable.

---

> ### Author Response · Authors · 2020-11-20
> **Response to AnonReviewer3 (Part 1)**
>
> Thanks for your constructive review comments. Below are our responses to your comments.
>
> **[Whether block-wise search is also used in student training and thus lead to unfair comparison with baselines]**
>
> Our pipeline consists of search stage and evaluation stage. The block-wise distillation is only conducted in the search stage to reduce the search space and improve the search efficiency. In the evaluation stage, the NAS-BERT model searched by our algorithm is pre-trained from scratch (without inheriting weights from the supernet) and then fine-tuned for a fair comparison with baseline. Thus, the improvement of our NAS-BERT comes from the discovered architectures.
>
> We also conduct experiments on NAS-BERT and the baseline using block-wise distillation in the pre-training in evaluation stage. We split the student model into different blocks following Fig. 2 in the paper. Each block is trained under the supervision of the corresponding teacher block. Since block-wise distillation does not jointly optimize different blocks, we also use standard distillation as used in NAS-BERT in the second half of training steps. The results are shown as follows:
>
> | Model          |\|| MNLI |\||  QQP |\|| QNLI |\|| CoLA |\|| SST-2 |\|| STS-B |\||  RTE |\|| MRPC |\||  AVG |
> |-------------------|:-:|:----:|:-:|:----:|:-:|:----:|:-:|:----:|:-:|:-----:|:-:|:-----:|:-:|:----:|:-:|:----:|:-:|:----:|
> | $\text{BERT}_{60}$          |\|| 83.2 |\|| 90.5 |\|| 90.2 |\|| 56.3 |\||  91.8 |\||  88.8 |\|| 78.5 |\|| 88.5 |\|| 83.5 |
> | $\text{NAS-BERT}_{60}$        |\|| 84.1 |\||  91.0  |\|| 91.3 |\|| 58.1 |\||  92.1 |\||  89.4 |\|| 79.2 |\|| 88.5 |\|| **84.2** |
> | $\text{BERT}_{60}+\text{Block}$    |\|| 82.2 |\|| 90.4 |\||  90.0  |\|| 54.4 |\||   92.0  |\||  88.7 |\|| 74.9 |\|| 87.7 |\|| 82.5 |
> | $\text{NAS-BERT}_{60}+\text{Block}$|\|| 83.1 |\|| 90.8 |\|| 90.3 |\|| 54.8 |\||  92.2 |\||  89.2 |\||  77.0  |\||  87.0  |\|| **83.1** |
>
> **Title**: Comparison between block-wise distillation and two-stage distillation. "+Block" means that the student model is trained with half of the training steps and continues to be trained with distillation as used in NAS-BERT with the remaining training steps.
>
> We find that block-wise distillation performs worse than the normal distillation in both BERT baseline and NAS-BERT. In the block-wise distillation setting, NAS-BERT still outperforms BERT baseline.

---

> > ### Author Response · Authors · 2020-11-20
> > **Response to AnonReviewer3 (Part 2)**
> >
> > **[About the novelty]**
> >
> > We respectfully disagree with your comment on novelty and would like to emphasize that our method is not simply a combination of existing ideas.
> >
> > - The key challenge for task-agnostic and adaptive-size BERT compression is computational complexity, as both neural architecture search and pre-training are extremely costly. Our main innovation is to reduce complexity for neural architecture search in pre-training, and we introduce three techniques: block-wise search, progressive shrinking, and performance approximation, to reduce the training cost, improve search efficiency and achieve strong experiment results.
> > - Note that besides block-wise search, our newly proposed progressive shrinking and performance approximation are critical for efficiency and effectiveness. Specifically, for progressive shrinking, we split the models into different bins to prevent the smaller models from being pruned and guarantee that we can get various architectures to meet different constraints.
> > - Even if for block-wise search, our method has the following key differences and main advantages compared with block-wise search. We add identity operation to allow an adaptive depth of searched architectures, which is not considered in [1], and thus [1] cannot automatically search models with adaptive depths. [1] solves this problem by manually training different depths to get several models, which is acceptable in lightweight image tasks, but is extremely costly in BERT pre-training task and cannot support devices with different kinds of memory and latency limitations.
> > - Once-for-all is a general idea to train a supernet with multiple architectures, and different works adopt different methods to achieve this purpose. For original once-for-all [2] and BigNAS [3], they rely on special designs (elastic kernel size, etc.) to train an adaptive network, which cannot explore novel architectures.  However, NAS-BERT can search for novel architectures. Specifically, we design search space including MHA, FFN, and especially the lightweight and efficient separable convolution operation, which demonstrate to be very helpful for BERT compression.
> >
> > **[About whether NAS-BERT is a good way to obtain the desired task-agnostic compressed models]**
> >
> > - For the first comment: the excellence of these models are tied to the specific set of shared parameters obtained. Note that after obtaining the searched model, we pre-train it from scratch without inheriting weight from the supernet and thus the excellence of NAS-BERT is not tied to the specific set of shared parameters.
> > - For the second comment: the excellence of these models is tied to the pre-training task. Note that the pre-training task in BERT [4] is designed to learn general representation, which can generalize well on downstream language understanding tasks, and is downstream task agnostic. The search of NAS-BERT is conducted on the pre-training stage. As long as a downstream task is well supported by BERT, it can be also well supported by NAS-BERT. Therefore, our NAS-BERT is task-agnostic from this point. The results in Table 2 and Table 3 in the paper demonstrate the effectiveness of NAS-BERT on GLUE tasks.

---

> > > ### Author Response · Authors · 2020-11-20
> > > **Response to AnonReviewer3 (Part 3)**
> > >
> > > **[About the fair comparison with previous works]**
> > >
> > > In Table 2 in the paper, we fairly compare NAS-BERT with the BERT baselines with exactly the same training configurations, which demonstrates the effectiveness of our method. Yes, as you say, when compared with previous work, there are many implementation differences, but we try our best to reduce the impact of these differences for fair comparison as follows:
> > > - Different teacher models. We have already compared our teacher models with those used in other works in Table 6 in the Appendix A.2 of the paper and clarified this kind of unfairness in the submitted version. Even without considering RTE and CoLA on which our teacher model achieves better accuracy, NAS-BERT still outperforms existing works as shown in Table 3.
> > > - Batch size and training steps. For the student training, DistilBERT uses a larger batch size (4096 batch size * 62,500 steps) and MiniLM uses more computations (1024 batch size * 400,000 steps). We pre-train the NAS-BERT from scratch with computations (2048 batch size * 125,000 steps), which has the same computations as BERT and most of the related works (256 batch size * 1,000,000 steps). Although MiniLM uses more computations, NAS-BERT still outperforms MiniLM in the 60M model setting.
> > > - Sophisticated distillation techniques. Hidden layer distillation (DynaBERT, TinyBERT, etc), attention score distillation (DynaBERT, TinyBERT), probe classifiers (AdaBERT) can be used to boost the performance. These techniques are efficient and are complementary to NAS-BERT. Our work mainly focuses on finding efficient and lightweight models for task-agnostic and adaptive-size BERT compression, but not aims to incorporate each sophisticated technique to train the searched architectures.
> > >
> > > We try our best to make the comparison with other works as fair as possible, without using sophisticated distillation skills, more computations, or a much better teacher. Even without sophisticated distillation techniques, NAS-BERT outperforms previous works as shown in Table 3 in the paper, which demonstrates the effectiveness of our searched architectures.
> > >
> > >
> > > [1] Li C, Peng J, Yuan L, et al. Block-wisely Supervised Neural Architecture Search with Knowledge Distillation[C]//Proceedings of the IEEE/CVF Conference on Computer Vision and Pattern Recognition. 2020: 1989-1998.
> > > [2] Cai H, Gan C, Wang T, et al. Once-for-All: Train One Network and Specialize it for Efficient Deployment[C]//International Conference on Learning Representations. 2019.
> > > [3] Yu J, Jin P, Liu H, et al. Bignas: Scaling up neural architecture search with big single-stage models[J]. arXiv preprint arXiv:2003.11142, 2020.
> > > [4] Devlin J, Chang M W, Lee K, et al. BERT: Pre-training of Deep Bidirectional Transformers for Language Understanding[C]//NAACL-HLT (1). 2019.

---

### Decision · Program_Chairs · 2021-01-07
**Final Decision**

**Decision:**

Reject

**Comment:**

Compressing BERT is a practically important research direction. Our main concern on this submission is on its practical value. Comparing with MobileBERT in the literature, NAS-BERT does not show advantages on any aspect: latency, prediction performance, or model size (less important), while being much more costly to build because of NAS. MobileBERT just simply narrowed the original BERT models (8x narrower than BERT large). So it is hard to convince the readers that adaptive-size or NAS is interesting or matters. On the research side, this paper have some interesting points on designing the search space, but overall the novelty of this paper is limited, as all of the reviewers pointed out. It is also worth noticing that the claim of "task agonistic" in this paper does not fully hold: in the downstream tasks, the soft labels of the teacher model are required to train the compressed model. To be fully "task agonistic", the results on downstream tasks should be solely based on training with the ground truth labels, as in the MobileBERT paper. Once following the exact task agnostic experimental protocol, the reported performance in this paper may be significantly lower.